# Determining protein structures in cellular lamella at pseudo-atomic resolution by GisSPA

Jing Cheng [1,5], Tong Liu[2,5], Xin You [3], Fa Zhang[4], Sen-Fang Sui [3], Xiaohua Wan [4,6] ✉ & Xinzheng Zhang [1,6] ✉

Cryo-electron tomography is a major tool used to study the structure of protein complexes in situ. However, the throughput of tilt-series image data collection is still quite low. Here, we show that GisSPA, a GPU accelerated program, can translationally and rotationally localize the target protein complex in cellular lamellae, as prepared with a focused ion beam, using single cryo-electron microscopy images without tilt-series, and reconstruct the protein complex at near-atomic resolution. GisSPA allows high-throughput data collection without the acquisition of tilt-series images and reconstruction of the tomogram, which is essential for high-resolution reconstruction of asymmetric or low-symmetry protein complexes. We demonstrate the power of GisSPA with 3.4-Å and 3.9-Å resolutions of resolving phycobilisome and tetrameric photosystem II complex structures in cellular lamellae, respectively. In this work, we present GisSPA as a practical tool that facilitates high-resolution in situ protein structure determination.

After decades of development, single-particle cryo-electron microscopy (cryo-EM) has determined the structure of proteins in solution to near-atomic resolution or even to atomic resolution on apoferritin[1,2]. However, proteins in their working state cellular environments often have more native conformations and form more complete complexes than purified proteins. Thus, solving the structure of proteins in situ is an important objective of cryo-EM.

Cryo-electron tomography (cryo-ET) is the most popular tool to study protein structure in situ[3,4]. In cryo-ET, tilt-series images are collected to reconstruct a tomogram. Many copies of identical protein complexes in tomograms are averaged to improve the resolution of the structure in situ. To increase the resolution, various software packages[5–11] have been developed to improve the quality of tomograms and the averaging procedure. Mini cells[12,13] and *Mycoplasma pneumoniae* (*M. pneumoniae*) cells[8,14–16] that grow on a grid have ~150-nm-thick regions that can be imaged directly by cryo-EM. However,

most cells are too thick for transmission electron microscope imaging. Thus, focused ion beam (FIB)[17,18] has been used to mill frozen cells to produce lamellae thin enough (150–200 nm) to be imaged with cryo-EM. But the thinning procedure has been reported to cause stress in lamellae[19–21]. The poor conductivity of lamellae and the thinning-induced stress can lead to severe beam-induced motion during imaging[19,20,22]. FIB milling can also cause radiation damage and ion contamination on the lamellae surfaces. Thus, it is challenging to solve the structure of proteins at high resolution in lamellae.

Although the collection of tilt-series images can be accelerated by beam-shift data collection[23–25], the cryo-ET throughput is still ~1/30th that of single-particle data collection[23]. Thus, the data collection throughput in tomography is an unsolved problem for high-resolution structures. Furthermore, tomogram deformation caused by beam-induced motion is a major factor that limits the reconstruction resolution[8]. Multi-particle system modeling has been shown to correct

[1]National Laboratory of Biomacromolecules, CAS Center for Excellence in Biomacromolecules, Institute of Biophysics, Chinese Academy of Sciences, Beijing 100101, China. [2]High Performance Computer Research Center, Institute of Computing Technology, Chinese Academy of Sciences, Beijing 100190, China. [3]State Key Laboratory of Membrane Biology, Beijing Advanced Innovation Center for Structural Biology, School of Life Sciences, Tsinghua University, Beijing 100084, China. [4]Beijing Institute of Technology, Beijing 100081, China. [5]These authors contributed equally: Jing Cheng, Tong Liu. [6]These authors jointly supervised this work: Xiaohua Wan, Xinzheng Zhang. ✉e-mail: wanxiaohua@bit.edu.cn; xzzhang@ibp.ac.cn

the deformation and effectively improve the resolution[8]. However, this requires high-abundance proteins with identical conformations as markers, or, alternatively, co-refining with high-abundance proteins to correct the deformation. This is a limitation.

In contrast to three-dimensional (3D) tomograms, proteins overlap with other biomacromolecules in two-dimensional (2D) cryo-EM images. To solve in situ structures directly from these 2D images without tomogram and tilt-series images, the target proteins have to be translationally and rotationally localized in a density-overlapped cryo-EM image. Then the structure has to be locally refined within the overlapping densities, which has been shown to be valid with a number of programs[5,7,8,11,26,27]. Rickgauer et al. used a high-resolution reference and a whitening filter to localize target proteins within overlapping densities[28]. However, they did not provide a complete workflow to obtain a high-resolution structure from the localized particles with overlapping densities. We developed the isSPA method[27], which was a complete workflow for solving protein structures with overlapping densities directly from cryo-EM 2D images. For better localization of the target protein, a signal-to-noise ratio (SNR) weighting function was deduced to calculate cross-correlation by using overlapping densities as noise. We also designed a sorting method included in isSPA to remove false particles and reduce the model bias effect. With this workflow, we successfully solved high-resolution protein structures in non-cryo-sectioned samples, such as glycoproteins on virus particles, a protein complex in subcellular organelles, and a membrane protein embedded in liposomes. Lucas et al. also implemented a complete workflow (2DTM) in *cis*TEM and calculated the structure of ribosomes in vitrified cells with a reported resolution of 4.3 Å[15]. However, they stated that the estimated resolution was unrealistically high because of the well-known effect of template bias, and, therefore, they displayed a low-pass filtered structure to 20 Å. The lack of successful applications of cryo-sectioned cellular samples indicated that the global localization of proteins and the refinement of the structure on FIB-milled lamellae need to be further explored.

In this work, we optimize the isSPA method with graphics-processing-unit (GPU) acceleration (GisSPA), which greatly improves the calculation efficiency. We examine the feasibility of applying GisSPA to FIB-thinned *Porphyridium purpureum* (*P. purpureum*) cellular lamellae. Without tilt-series images, we solve the phycobilisome (PBS) (14.7 million Dalton (MD)) and the tetrameric photosystem II (PSII) complex (1.5 MD) with 3.4 Å and 3.9 Å resolutions, respectively. Thus, GisSPA is practical and efficient for high-resolution structure determination on FIB-thinned lamellae. We also show that the localization efficiency depends on the lamella thickness and the molecular weight of the target protein. The lower limit of the protein molecular weight when using GisSPA on cellular lamellae is ~1.1 MD, which covers many important protein complexes and covers almost all the protein complexes that have been reported beyond 10-Å resolution in cellular lamellae via cryo-ET[22,29–31].

## Results
### isSPA and GPU acceleration
Localization of a target protein within overlapping densities is based on SNR-weighted projection matching via a cross-correlated gram (CCG) calculation. The output parameter is the ratio of the peak to the standard deviation of the CCG, which is used to decide whether to keep a location. The weighted cross-correlation is defined as

$$cc = \sum_k W(k) \cdot X(k) \cdot M^*(k) \qquad (1)$$

where, $k$ is the spatial frequency, $M^*(k)$ is the complex conjugate of the Fourier transform of a projection of a 3D template, $X(k)$ is an image in Fourier form, and $W(k)$ is the optimized weighting function that was

derived as[27]

$$W(k) = \frac{CTF(k) \cdot FSC(k)}{\frac{1}{SSNR(k)} + n \cdot CTF^2(k)} \qquad (2)$$

CTF is the contrast transfer function without the envelope function from the cryo-EM image, SSNR is the ratio of target protein density to shot noise evaluated from the power spectrum, FSC is the gold-standard Fourier shell correlation of the 3D template, and $n$ is the ratio of overlapping protein intensity to target protein density. An increase in $n$ indicates more noise from overlapping protein densities, and the normalized high-frequency weight increases with $n$.

The CCG calculation described above is computationally demanding and images are often binned to reduce the computation time. For example, the localization task of 2,295 projections on a set of 1437 images (2.5 K × 2 K) required ~60 days using 200 cores (E5-2660 CPU) in parallel. The optimization and GPU acceleration (Fig. 1a) of the algorithm enabled the same task to be completed within three days with seven GTX 1080 GPUs running in parallel. The computational efficiency was increased by factors ranging over 500–900 for different types of GPUs (Fig. 1b).

To solve the memory-bound problem, the whole image was split into several small windows that share the same templates. The small windows save memory, as shown in Fig. 1c, which expanded the type of GPUs able to run the task. Neighboring windows had an overlapped region larger than that of a target protein, which ensured that any target protein in the micrograph was complete in at least one window. When the window size was nearly twice the overlap or smaller, the overlap between windows sharply increased the occupied memory. This occurred only when the target protein was extremely large. The GisSPA program was user-friendly and only one command line was required to complete target protein localization after proper parameters were set in the configuration file.

### Localization of proteins on cellular lamellae
To test the ability of GisSPA to localize proteins on cellular lamellae, *P. purpureum* cells were frozen on holy-carbon copper grid and thinned with FIB. The lamellae were immediately imaged with a 300 kV cryo-electron microscope (Titan Krios) equipped with a K3 direct camera operated in super-resolution mode. We acquired 1437 energy filtered images (20-eV slit width) with a physical pixel size of 1.668 Å and a 6.4-s exposure time, leading to a total dose of 35 e$^-$/Å$^2$ fragmented into 80 frames. We estimated the thickness of the lamellae in the 1437 images from the ratio of elastic electrons to the total number of electrons (see "Methods" for details). The calculated thickness distribution indicated that 95% of the lamellae were 100–350 nm thick and that 50% of the lamellae were thicker than 200 nm (Supplementary Fig. 1).

A 2.8 Å-resolution 3D single-particle cryo-EM map (EMD-9976) of PBS[32] was used as the high-resolution reference (Fig. 2a) to localize PBSs in all the micrographs. In total, 2295 2D projections (ignoring in-plane rotations) were generated from the 3D map using an angular step of three degrees. An optimized CCG was calculated between each projection and micrograph as described above. Only frequencies in the range from 1/100 Å$^{-1}$ to 1/9 Å$^{-1}$ were used in the calculation and the overlapping parameter $n$ was set to 3. The cutting threshold of the output score was set to 8.

### Refinement of the PBS-PSII complex
Based on the localization results (Fig. 2b), we obtained 325,892 PBS particles from 1437 micrographs that met the cutting threshold. Then, non-alignment 3D classification was performed to divide the particles into three classes. All the class averages featured double-layer thyla-koid membranes (Supplementary Fig. 2). Because the reference map did not contain this feature, false detections were limited at the threshold. Therefore, all the particles were retained for further data

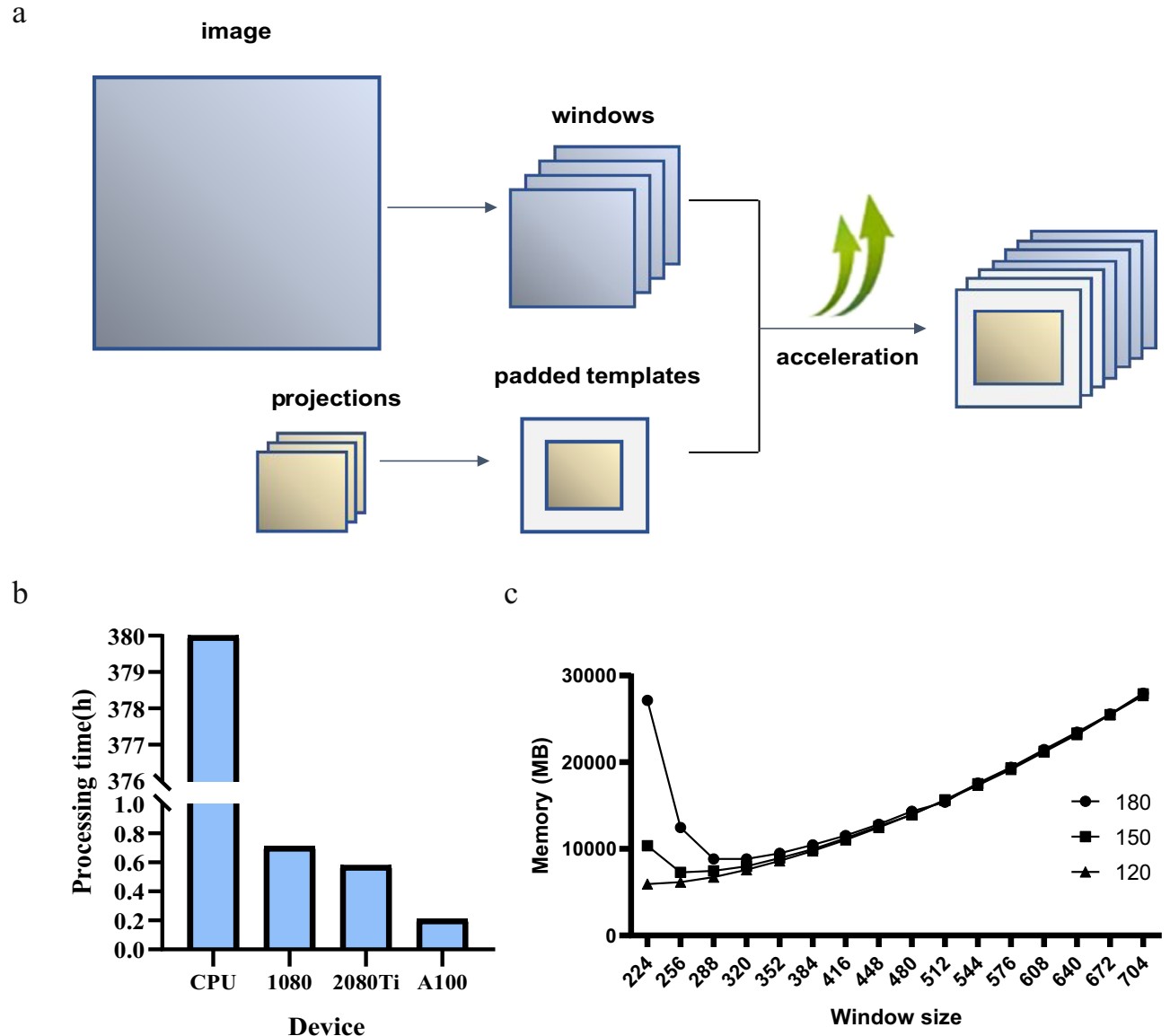

**Fig. 1 | GPU acceleration and optimization. a** Padding method with splitting image into windows. The operation of splitting reduces the number of padded zeros for CCG calculation, which is accelerated by GPU. **b** Processing time of 2295 projections matching with a 5 K × 4 K image on CPU and different types of GPUs. **c** Occupied memory with different window sizes and overlap settings (180, 150, and 120).

processing after 3D classification. The 3D classification also revealed distinct densities connecting to PBS at the thylakoid side that may be attributed to PSIIs. We then used local refinement to increase the resolution of the PBS−PSII complex. The starting translational and rotational parameters resulted from of the localization. The PSII densities in the refined structure were of poor quality. Therefore, we performed a round of 3D classification focused on the PSII region of the complex without alignment via RELION[33]. A total of 251,458 particles were selected in the 3D classification, which contained three PSII dimers in addition to PBS. Approximately 23% of the particles were excluded because of the different ways PSII connecting to PBS. For example, some of the excluded PSIIs were connected to PBS in a more flexible manner than other PSIIs, possibly because they were located in a twisted membrane. Local refinement of the PBS−PSII complex led to a 3.4 Å-resolution reconstruction at the PBS region (Fig. 3c, d). Features typical for this resolution range, such as amino-acid side-chain stubs and β-strands as shown in Fig. 2e, are observed in the map. The mean displacement of PBS particle positions after refinements is about 1.1

angstroms and the mean Euler displacement is 1.4 degrees (Supplementary Fig. 3). Focused classification and refinement were performed on three PSII dimers in contact with the PBS core, resulting in three reconstructions of PSII dimers with 3.6 Å, 3.7 Å, and 4.4 Å resolutions (Fig. 2f, g).

Lowering the threshold of the score during the PBS localization in cryo-EM micrographs could identify more PBSs, but increase the portion of false particles and caused model bias problems. Because only frequencies from 1/100 Å$^{-1}$ to 1/9 Å$^{-1}$ were used in the calculation, false particles would cause a sudden drop in the Fourier shell correlation (FSC) at frequencies >1/9 Å$^{-1}$[27], which was not observed in our FSC curve. Furthermore, the 3D model used in the SNR-weighted projection matching was from purified PBS. Our in situ PBS structure[34] had extra densities that were missing or flexible in the purified PBS. The contour level of the extra densities was similar to those of nearby densities (Fig. 2h), indicating that almost all the particles were true detections at this threshold. On this basis, we used the PBS localization result as a positive control for the localization task described below.

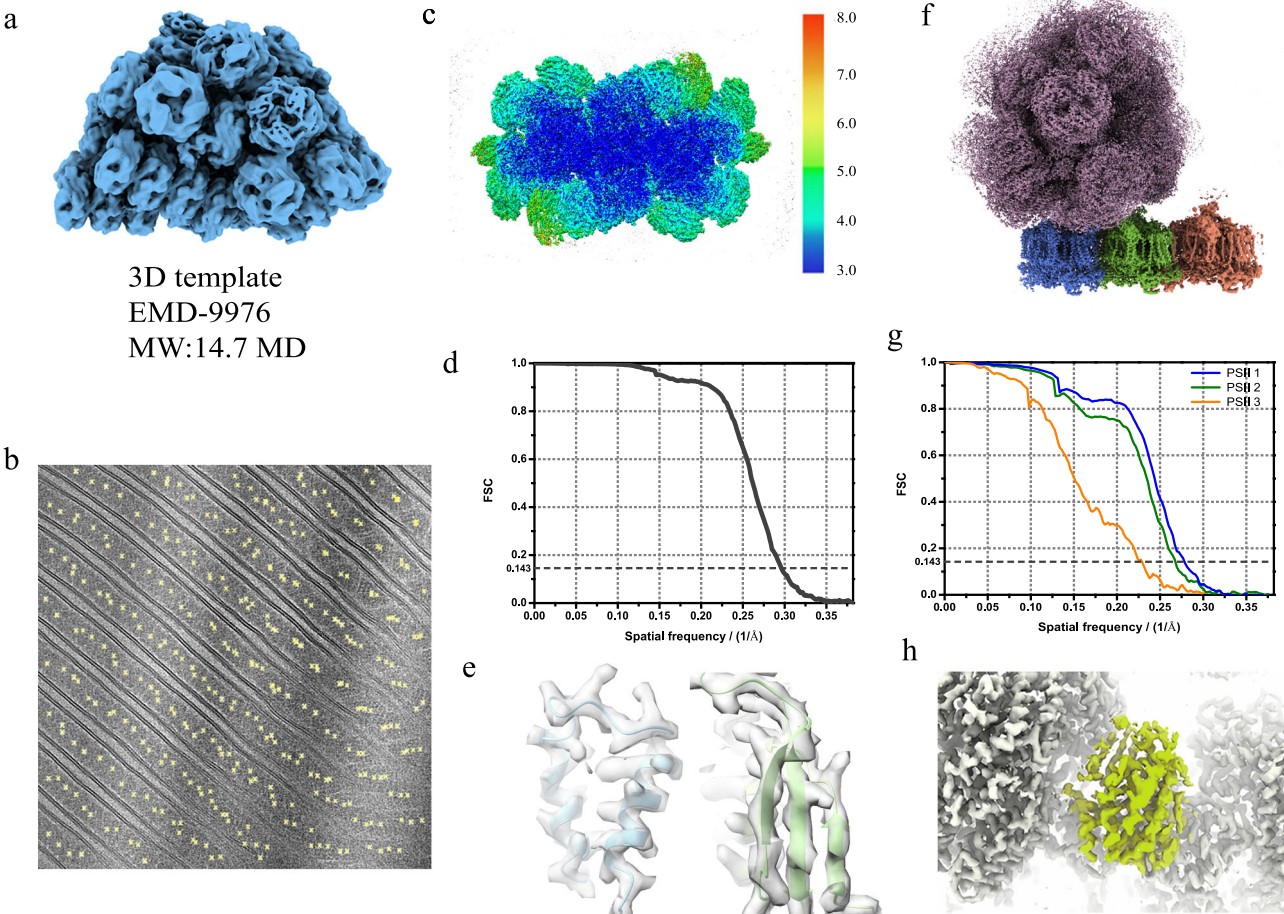

**Fig. 2 | Localization and high-resolution reconstruction of PBS-PSII complex.**
**a** 3D PBS model (EMD-9976) used in the projection matching, and the molecular weight is about 14.7 MD. **b** Locations on the image from projection matching at threshold 8. **c** Isosurface of the 3.4 Å-resolution PBS map colored by local resolution, showing a high resolution near the core and a lower resolution outside may due to the flexibility. **d** The FSC curve of PBS structure. **e** High-resolution features of PBS map fitted by pdb 6kjx, such as amino-acid side chains (left) and β-strands (right), are clearly identified for this resolution range. **f** Overall structure of PBS-PSII complex, with 3 PSII dimers (PSII 1 in blue, PSII 2 in green and PSII 3 in orange) connected to the PBS. **g** FSC curves of three PSII dimers, indicating the resolutions of 3.6 Å (blue, PSII 1), 3.7 Å (green, PSII 2) and 4.4 Å (orange, PSII 3) at FSC 0.143. **h** Extra densities in PBS map (colored in light green). Source data are provided as a Source Data file.

## Performance of GisSPA on cellular lamellae

In thick lamellae, inelastic scattering and multi-elastic scattering reduce the signal from the target protein. Overlapping densities from other biomacromolecules also add more noise to the cryo-EM image, which further reduces the SNR. Thus, the power of localizing a target protein on lamella may be negatively related to the thickness of the lamella. To test the detection efficiency of GisSPA on lamellae with different thicknesses, we used the rotation and translation results of PBS described above as the positive control. Micrographs were divided into four groups based on their thicknesses (Fig. 3a). A 1.6-MD part of PBS was used as the target, which was localized in micrographs of the different groups. Detection within a 2-pixel (~6.7 angstroms) transla-tional distance and 6-degree Euler distance relative to the control were considered true. Our results showed that the detection efficiency of GisSPA decreased sharply with increased lamella thickness when the thickness was >170 nm (Fig. 3a). Therefore, lamella thinning is a key procedure, and high-quality thin lamellae are important in GisSPA applications.

The molecular weight of target proteins is an important factor that affects detection efficiency. To test the lower weight limit that could be localized, we used different PBS regions with weights in the range 1.1–5.2 MD as targets (Supplementary Fig. 4), and calculated the detection efficiency of these targets in the micrograph group with

thicknesses of 110–125 nm (Fig. 3b). The GisSPA detection efficiency decreased sharply for the low molecular weights, which is consistent with previous results of non-lamella samples[27]. Our previous results indicated that at 10% precision, the 3D classification and the sorting method efficiently removed false particles, resulting in a reconstruc-tion via subsequent 3D-structure refinement. The recall of a 1.1 MD protein in lamellae was ~10% at 10% precision. For smaller proteins, the recall value was too low to determine the in situ structure in the tested lamellae. Thus, 1.1 MD was close to the lower limit of the protein molecular weight that could be solved on cellular lamella with GisSPA.

## Localization and reconstruction of the PSII complex

Having demonstrated that a 1.1 MD protein could be localized with GisSPA, we aimed to reconstruct the 1.1 MD protein in situ. However, because that the 1.1 MD protein was part of PBS, the continuous den-sities nearby greatly reduced the difficulties in the subsequent classifi-cation and refinement steps. Instead, a 1.5-MD tetrameric PSII complex was used as the target. The interaction between the PSII complex and other proteins was weak and could be disrupted with a soft mask that contained only the PSII complex for classification and refinement. The PSII complex structure was extracted from the 3.7 Å reconstruction as a 3D template, and 1,649 projections (ignoring in-plane rotation) were generated with an angular step of 5 degrees without applying

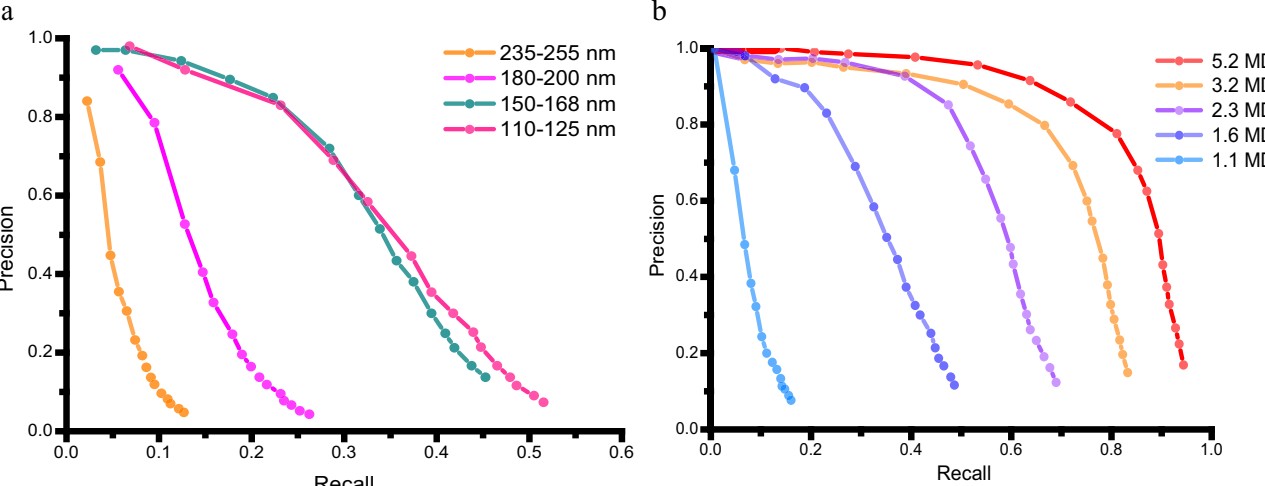

**Fig. 3 | Precision-recall curves. a** Precision-recall curves of the localization on images with a thickness range of 110 to 125 nm (pink), 150–168 nm (olive), 180–200 nm (magenta) and 235–255 nm (orange). The target protein is 1.6 MD in molecular weight. **b** Precision-recall curves of the localization of target proteins in 5.2 MD (red), 3.2 MD (orange), 2.3 MD (light purple), 1.6 MD (dark purple) and 1.1 MD (blue) on images with a thickness range of 110 to 125 nm. Source data are provided as a Source Data file.

symmetry. Data in the frequency range from $1/100\,\text{Å}^{-1}$ to $1/9\,\text{Å}^{-1}$ were used in the SNR-weighted projection matching. The threshold was 7.5, and 482,888 potential particles were extracted and subjected to non-alignment 3D classification with RELION[33,35]. A total of 98,603 particles were selected for subsequent local refinement with C2 symmetry applied, yielding a 5.7 Å-resolution reconstruction. The FSC curve before sorting exhibited a sharp drop close to 9 Å (Fig. 4a), indicating that the reconstruction was affected by model bias caused by false particles. To reduce the number of false particles, we performed sorting method on these particles[27] and 25,139 of them were selected. The FSC curve of half maps reconstructed from the removed particles dropped sharply to 0 near $1/9\,\text{Å}^{-1}$, while the FSC curve of the selected particles showed a wide range with relatively high FSC values beyond $1/9\,\text{Å}^{-1}$ (Supplementary Fig. 5a), which suggested that the removed false particles didn't produce bias at frequencies larger than $1/9\,\text{Å}^{-1}$. This phenomenon agreed with our previous finding[27]. Then we performed another round of local refinement, yielding a resolution of 4.7 Å. The density quality also had an improvement as showed in Supplementary Fig. 5b. This indicated that false particles led to inaccurate alignment during refinement and thus limited resolution. By adding 377 more micrographs, 33,660 more particles were joined. After another round of local alignment and sorting, only 23,410 particles were picked for the final reconstruction. The refinement was re-processed using a calibrated pixel size of 1.632 Å at this magnification. We obtained a 3.9-Å resolution reconstruction for the PSII complex (Fig. 4b and Supplementary Fig. 5c).

Model bias effect occurred in 2D projection matching was reported by Bronwyn et al.[15] for ribosomes in *M. pneumoniae* cells, where the resolution was unrealistically high. We performed SNR-weighted projection matching on their dataset (EMPIAR-10731)[15] with 50S ribosomes segmented from the cryo-EM map (EMD-11999)[8] as the 3D template at frequencies from $1/100\,\text{Å}^{-1}$ to $1/8\,\text{Å}^{-1}$. A total of 6675 potential particles were picked with a threshold 7.9. Using non-alignment 3D classification and focused local refinement, we obtained a 50S structure at 6.74 Å resolution with 4743 particles and a 30S structure at 8.93 Å resolution with 3939 particles (Fig. 4c). The ratio of 30S subunits to 50S subunits was 0.83, which was close to the tomography result (0.7) reported by O'Reilly et al.[14]. In addition, O'Reilly et al.[14] found that 30S exhibited multiple conformations. A relatively low particle number and multiple conformations may be reasons why the resolution of 30S is 2 Å worse than that of 50S. Densities of 50S and

30S were generally expected for the corresponding resolutions (Fig. 4c, d) and the FSC curves did not show a rapid drop at close to $1/8\,\text{Å}^{-1}$ (Supplementary Fig. 6). The 50S structure had extra densities to 50S model (Fig. 4e) as described by Bronwyn et al. Both these findings demonstrate that the overall 70S structure resolved by GisSPA was unbiased.

### Analysis on *P. purpureum* lamellae

Vitreous ice obtained by high cooling-rate freezing exhibits rapid motion at the initial stage of electron irritation. This induces both translational and rotational movement of individual proteins that cannot be corrected by motion correction software. Hence, there is reduced resolution on frame reconstructions initially exposed to 3–5 e⁻/Å². To examine whether this problem occurred on cellular lamellae, we calculated frame reconstructions from frames exposed to 1.32 e⁻/Å². The first six frame reconstructions had similar resolutions and were better than subsequent reconstructions (Fig. 5a). PBS is a large protein complex that enabled local refinement of rotational parameters in each frame reconstruction. This refinement produced a 0.4-Å improvement in the resolution of the first frame reconstruction. Therefore, only a mild rapid motion effect occurred on the cellular lamellae, which was different from single-particle data. This difference was likely attributed to our data collection methods.

In tomography, this rapid motion is presented as drum-like deformation, and ice deformation during imaging is the major issue that limits the resolution of sub-tomogram averaging[8]. In drum-like deformation, proteins move mostly along the direction normal to the lamella. In our data collection, the stage was tilted before data collection to compensate the pre-tilt of lamella. Thus, the motion caused by drum-like deformation is mostly along the direction of the incident electron beam. This reduced the beam-induced motion, eliminated the defocus gradient in the image, and facilitated the contrast transfer function (CTF) fitting.

The Rosenthal–Henderson plot[36] of PBS indicated that the B factor used to assess image quality was 131.1 Å². A previous single-particle reconstruction of the same PBS sample had a B factor of 122.7 Å² (Fig. 5b). These B factors were close, but the resolution of the single-particle reconstruction was higher than that of the lamella for same number of particles. To obtain a 3.5-Å resolution, in situ reconstruction on cellular lamellae needed ~111,723 particles, as indicated in the Rosenthal–Henderson plot[36], while only 47,083 were needed for

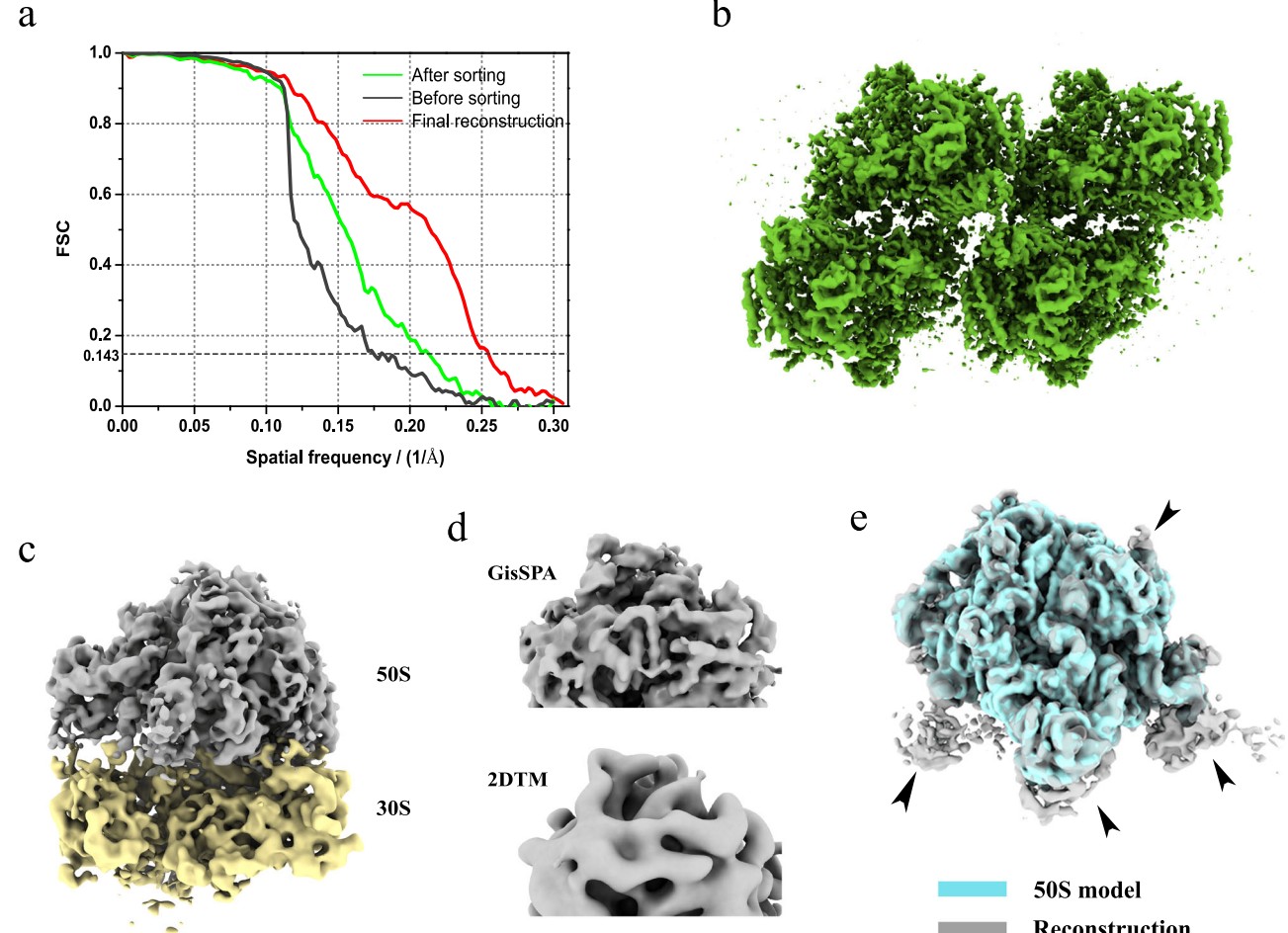

**Fig. 4 | PSII tetramer of *P. purpureum* lamella and ribosome of *M. pneumoniae* resolved by GisSPA. a** FSC curves of PSII tetramer reconstruction before (dark) and after (green) removing false positives by sorting method. The final resolution reached 3.9 Å after adding more micrographs (red). **b** Reconstruction of PSII tetramer at 3.9 Å. **c** 3D reconstruction resolved by GisSPA with a *M. pneumoniae* 50S as purified single particles. The main reason for the resolution difference may be that lamellae are much thicker than single-particle samples. 3D model, showing clear density for the 30S which is not included in the model. **d** Details of 50S map resolved by GisSPA (above) and 2DTM (below). **e** Comparison between 50S reconstruction by GisSPA (gray) and model (blue), showing differences indicated by arrows. Source data are provided as a Source Data file.

## Discussion

False particles produce bias effect in the reconstruction before the high-resolution cutoff in particle localization and reduces the reconstruction resolution during the GisSPA calculation. Hence, removing false particles is necessary for high-resolution structural determination. Due to the fact that the SNR is relatively low in frequencies higher than $1/8$ Å$^{-1}$ and that sorting method needs signals not included in the particle localization, signals in frequencies higher than $1/8$ Å$^{-1}$ are not suggested to be included in 2D projection matching.

Our results showed a mild rapid motion effect on the cellular lamella. The plunge freezing cooling rate of cells that are a few micrometers thick is lower than that for regular protein samples that are tens of nanometers thick. A lower cooling rate decreases stress in vitreous ice and therefore reduces the fast motion effect, as reported for purified protein samples[37–40]. Another possible reason is that the lamellae were radiated by FIB-milling ions and secondary electrons that may have partially released stress.

Instead of tilt series, GisSPA takes single images on lamellae. On a holey-carbon grid, data collection is performed in abundant holes with a thin layer of ice. Whereas, the number of good lamellae on a grid is limited and the entire thin area of the lamella is adequate for data collection because target proteins may locate anywhere. Thus, a fringe-free nanoprobe that is slightly larger than the camera could be used for data collection because an illumination beam that exceeds the camera size will irradiate the sample. Data collection throughput may be limited by the number of lamellae rather than the available time on the microscope. Thus, to collect as much area as possible on a lamella, spots for imaging should be planned carefully, keeping in mind that an area with double exposure can be distinguished from a good area by eye inspection as shown in Supplementary Fig. 7.

Block-based reconstruction[26] and GisSPA both perform structure refinement on cryo-EM images of target protein subject to overlapping densities from other biomolecules. State-of-art sub-tomogram averaging programs also refine the structure in 2D tilted images with overlapping densities[5,6,8,11]. Because the electron dose is split into each image in a tilt-series, the SNR of a single tilt image is usually too low to perform accurate refinements. Therefore, particle images that represent the same protein complex in a tilt-series are refined in a cooperative manner by adding restraints to the rotational and translational parameters based on the initial alignment of the tomogram. In GisSPA, a single image contains the entire electron dose, and therefore, refinement is relatively simple, and more effective and accurate than that in sub-tomogram averaging.

Certain membrane proteins with mostly transmembrane region in cellular environment are difficult to solve via STA, where outline of the protein is covered by continuous lipid layers, which is different from the soluble proteins. Thus, without correct low-resolution features, the

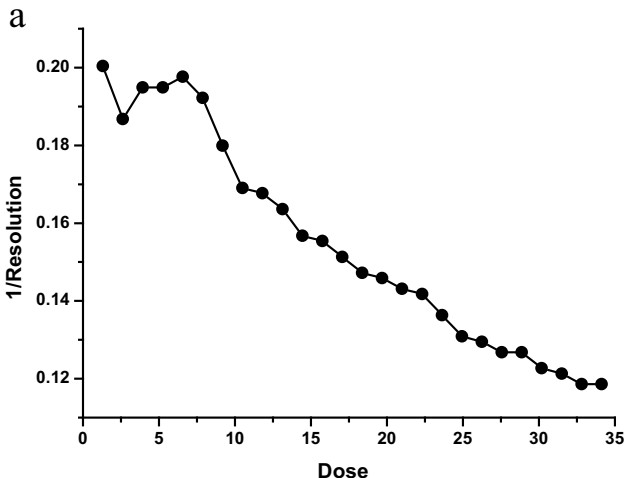

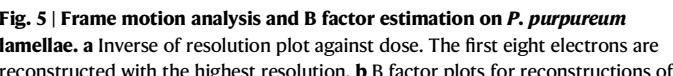

**Fig. 5 | Frame motion analysis and B factor estimation on *P. purpureum* lamellae. a** Inverse of resolution plot against dose. The first eight electrons are reconstructed with the highest resolution. **b** B factor plots for reconstructions of purified PBS (green) and in situ PBS (magenta) of *P. purpureum*, indicating similar B factor estimations. The error bars in both plots represent the standard deviation from eightfold random resampling. Source data are provided as a Source Data file.

traditional iterative reconstruction method may not work. A high-resolution template like we used for 2D localization can help to localize protein in 3D membranes in tomogram.

The B factor values of PBS in cellular lamella and purified proteins are close, probably because PBS is large and the decrease in SNR from the sample thickness barely affects the alignment. The alignment accuracy may get reduced when the target protein is reduced, indicating that smaller proteins have larger B-factor values than large proteins. In addition, identification precision is limited by the target protein weight and the sample thickness, thus, an improved milling method that can either increase the lamella quality or reduce the thickness and algorithms that can improve the precision are needed for an in situ structure determination with higher resolution.

## Methods

### Cryo-EM data collection and preprocessing
The lamellas of *P. purpureum* data were collected on a FEI Titan Krios EM operated at 300 kV equipped with a Gatan K3 Summit detector and an energy filter (20 eV slit width). For each lamella, the grid was tilted to either −15° or +15° before imaging, which making the lamella plane perpendicular to the incident beam. Data collection was performed with SerialEM[41] using a nominal magnification of 53,000 with a total dose of ~35 e$^-$ at super-resolution mode, resulting in a pixel size of 0.834 Å/pixel. A total of 1437 such movies were collected. The defocus value was set from 2.0 to 4.0 μm. Each micrograph was dose-fractionated to 80 frames. Motion correction was done by MotionCorr2[42] and micrographs were dose-weighted and binned with a factor of 2.

### Structure determination of PBS-PSII complex by GisSPA
The CTF parameters of micrographs were estimated by CTFfind4[43] and the contrast were inverted before performing SNR-weighted projection matching. To save memory, each 5 K × 4 K micrograph was binned by 2 and cropped into twelve 720 × 720 images with an overlapping area of 114 × 720. Due to the large size of the target protein, we set the size of the window to 512 and the overlapping area to 300 in pixels. The localization task was performed by seven GTX1080 GPUs in parallel for about 3 days, and locations with output score beyond 8 were selected as potential particles. To remove duplicated detections, potential particles within translational distance of 4 pixels (~13.3 angstroms) and Euler distance of 6 degrees were merged as one particle.

325,892 particles were 2-binned extracted by RELION with a box size of 220, after implementing a round of 3D classification using the 3D model in 2D projection matching as the initial model and the lowpass filter was set to 20 Å, we checked the 3D map by UCSF Chimera and all the particles were kept. Then the particles were re-extracted at bin 1 and local refined at an initial sampling of 1.8 degrees with the best reconstruction from 3D classification being the ab-initial model. Based on the refined particles, we performed a round of 3D classification focused on the PSII dimers region, 251,458 particles of good PSII density were selected in total. By local refinement with CTF parameters refinement and Bayesian polishing, we obtained a final resolution of 3.4 Å on PBS. In order to obtain a high resolution PSII structure, we expanded the particle symmetry to C1 and doubled the number of particles. Subsequent 3D classification was focused on PSII tetramer, 229,758 particles were selected and re-extracted by block-based methods to ensure the PSII being located in the center of image. Local resolution reached 3.6 Å in the PSII 1 and 3.7 Å in the PSII 2 dimers. The same process was done in the third PSII dimers, and 147,568 particles were kept to yield a resolution of 4.4 Å (Supplementary Table 1).

### Structure determination of PSII complex by GisSPA
Global localization of PSII tetramer was performed on the 720 × 720 images generated from the PBS detection. Potential particles within translational distance of 4 pixels (~13.3 angstroms) and Euler distance of 6 degrees were merged as one particle, resulting in 482,888 potential particles. Particles were un-binned extracted by RELION with a box size of 320 × 320, and divided in to ten classes without orientational and translational alignment. 143,685 particles in one class were kept and transferred to a second-round classification into five classes, and 98,603 particles were selected to subsequent local refinement.

The 2-fold axis of PSII tetramer was aligned by RELION and a map with C2 symmetry applied was generated. The rotations and translations between the original map and the newly generated map with C2 symmetry were applied to each particle.

Since the resolution of PSII tetramer has reached 5.7 Å based on 98,603 particles, we scored each particle with the sorting method at a frequency range of 1/9 Å$^{-1}$ to 1/5.7 Å$^{-1}$. Particles with absolute output score below 0.04 were removed, resulting in 25,139 particles for the reconstruction at a resolution of 4.7 Å.

To improve the resolution of PSII complex, we collected 377 more micrographs of *P. purpureum* lamellae, 2D projection matching and non-alignment 3D classification were applied same as described above. Then 33,660 particles were selected and put together with the 25,139 particles, a round of local refinement was applied and the resolution was reported as 4.2 Å. We performed another round of sorting on

these particles using frequencies range from $1/8\,Å^{-1}$ to $1/4.3\,Å^{-1}$, 23,410 particles were selected by a threshold of 0.015. Using the calibrated pixel size 1.632 Å and performing one round of CTF refinement, the resolution reached 4.0 Å by RELION. To further push the resolution, the 23,410 particles were transferred to cryoSPARC[44], yielding a final resolution of 3.9 Å by non-uniform refinement.

## Localization and reconstruction of Ribosome in *M. pneumoniae*
We downloaded 220 motion-corrected micrographs from https://www.ebi.ac.uk/empiar/EMPIAR-10731/, CTF parameters were estimated with CTFfind4[43]. Then we excluded fields containing gold fiducial beads, which can gather localizations, from micrographs. The 50S large subunit structure of *M. pneumoniae* (EMD-11999)[8] was used as the 3D model in SNR-weighted projection matching. The weak density of 30S was removed from the model before calculation. The overlapping parameter *n* was set to 2 and the output score threshold was 8, yielding 6675 particles. A round of 3D classification without alignment was performed on these particles, dividing in to two classes, and 4743 particles were selected to local refinement. We obtained a resolution of 6.7 Å in 50S large subunit region, however, the 30 S small unit showed worse density than 50S. It suggested that the 30S had lower occupancies than 50S, which has been investigated[14]. Therefor we performed another round of non-alignment 3D classification into two classes focused on the 30S region based on the refined particles, and 3,939 particles presenting features of 30S were kept. A focused local refinement on 30S led to a resolution of 8.9 Å.

## Ice thickness estimation
Sample thickness was estimated by the Beer-Lambert law[45,46], the value of electron dose per angstrom square was treated as the intensity over vacuum and the elastically scattered electrons counts passing through sample and energy filter as the intensity over sample. The counts of elastic electrons are compensated by 22% due to the presence of coincidence loss[47,48]. Based on the averaged thickness (150–200 nm) of *P. purpureum* studied by previous tomography[49], an estimation of 300 nm was set as the mean free path for electron scattering in lamellae.

## B factor estimation
Particles in two random halves from the auto-refinement in RELION were split into eight subsets of 660, 1100, 1800, 3000, 5000, 8000, and 13,000 particles. The orientations from the 3.4 Å-resolution reconstruction were used to calculate reconstructions. For each subset and repeat, resolution was estimated using standard post-processing functionality in RELION with a same mask. The inverse square of the resulting estimated resolution was then plotted against the natural logarithm of the number of particles in the subset, and B factors were calculated from the slope of linear fitting of all points in the plot.

## Reporting summary
Further information on research design is available in the Nature Portfolio Reporting Summary linked to this article.

## Data availability
Cryo-EM density map of PSII complex generated in this study has been deposited in the EM Data Bank under the accession code EMD-35045. Source data are provided with this paper.

## Code availability
Source code, and user guide of GisSPA are available at https://github.com/Autumn1998/GisSPA.

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

## Acknowledgements

We thank L. Kong for cryo-EM data storage and backup. The project was funded by the National Key R&D Program of China (2021YFA1301501 and 2017YFA0504700 to X.-Z.Z.), the National Natural Science Foundation of China (31930069 and 32150010 to X.-Z.Z., 61932018 and 32241027 to F.Z., 62072441 to X.-H.W.), the Strategic Priority Research Program of the Chinese Academy of Sciences (XDB37040101 to X.-Z.Z.) and the Key Research Program of Frontier Sciences at the Chinese Academy of Sciences (ZDBS-LY-SM003 to X.-Z.Z.).

## Author contributions

X.-Z.Z. supervised the project. S.-F.S. and X.Y. provided the *P. purpureum* lamellae cryo-EM data. T.L. and X.-H.W. wrote the codes for GPU acceleration. X.-H.W. and F.Z. supervised the code writing. J.C. solved the structures of PBS and PSIIs and conducted the analysis. X.-Z.Z., J.C., and T.L. wrote the initial draft, and all authors contributed to the discussion of the results and revision of the manuscript.

## Competing interests

The authors declare no competing interests.
