## [Peer Review File · Nature Communications]

Determining protein structures in cellular lamella at pseudo-atomic resolution by GisSPAREVIEWER COMMENTS

Reviewer #1 (Remarks to the Author):

The manuscript from Cheng et al describes a new package for 2D template matching. The novelty, in this case, is lacking as the method was described already. Nevertheless, the GPU acceleration is worth attention as the previous implementations of this method requires multiple days to process a single micrograph.

The article is well written and easy to understand.

In order to better follow whether the work is worth the novelty, I would like to see a proper comparison with the existing methods (e.g. process the same data using CISTem and compare the difference in accuracy). Further, by extracting the particles with no randomisation of the angles, the template bias will be inevitable, and the resolution reported may be effectively an over-estimation.

In principle, the experiment is easy to perform.

To properly review the article, I downloaded the software, which runs with minor bugs at this stage, and I have noticed a difference in performance compared to CISTem, making it faster, but data displaying more charging is also less accurate.

An in depth analysis of the performance, accuracy and bias will definitely make the article worth publication.

Reviewer #2 (Remarks to the Author):

Review of Zhang et al. 2022, "Determining protein structures in cellular lamella at pseudo-atomic resolution by GisSPA"

In the manuscript, Zhang et al. present their GPU-accelerated software - GisSPA - for 2D template matching on cellular cryo-FIB lamellas. The authors present three examples of sub-nanometer structures – Phycobilisomes (*P. purpureum*, C2 symmetry, 3.4 Å), photosystem II complex (4.7 Å, C4), and

ribosomes (6.9-8.9 Å, *M. pneumoniae*, C1) – reconstructed from their hits and use a special sorting algorithm to reduce the template bias introduced by searching their projection images against high resolution reference structures.

In summary, the paper presents interesting results and an alternative to subtomogram averaging (STA) for targets that have indeed been more difficult to resolve at high resolution by STA. While the method for sure could be interesting for some, overall it is questionable that novelty, applicability, and efficiency make GisSPA and its manuscript suitable for Nature Communications.

Major:

- Novelty

As far as this reviewer can tell, nothing major has changed from isSPA, except for the GPU-acceleration. Admittedly, this is a major improvement and for sure a lot of work has gone into that. While this has reduced the time needed for performing the template search from months (60 days for 200 CPUs, line 124) to days, there is still a significant commitment of resources necessary to perform the template matching task.

Additionally, a similar approach has already been introduced by Grigorieff et al.

- Efficiency

As mentioned before, this reviewer recognizes the improvement due to GPU acceleration, but running the software on seven GTX 1080 GPUs over three days is still a major time/resource investment hardly compatible with “screening” different targets and “standard” single-particle like workflows. Overall, this will limit the applicability of the GisSPA to specific projects with clear targets.

- Constraints

- Lamella thickness

The authors rightfully point out that GisSPA requires relatively thin lamellas (<170 nm). Thickness of lamellas is however inversely proportional to their stability/quality. Producing such lamellas is more difficult and not as easily automated, limiting the sample preparation for GisSPA.

- Molecule weight

The authors suggest that the lower limit for detection is ~ 1.1 MDa. Of course, there is a limit for template matching and STA, too, however significantly smaller complexes such as Rubisco (~0.5 MDa)

have already been solved in situ (Engel B. et al.). This furthermore limits the general applicability of GisSPA.

- Model bias and claim of “bias free”

The model bias is mitigated by a “sorting algorithm” implemented already in isSPA, where “particles (...) are sorted according to their scores” (isSPA publication in The Innovation 2, 100166, November 28, 2021), which - unless this reviewer fails to see a more important step – seems like an oversimplification and stretch of the word “algorithm”. Even a bimodal distribution is not a sufficient criterion for selecting true and false positives since CCCs between template matching runs are not comparable (even when normalized). This is a misconception sometimes also presented in STA papers.

Minor:

- “therefore, rotating at the stage before data collection is operated to make sure the normal direction is along the direction of the incident electron beam and large protein motion is avoided.”

Do the authors mean that they account for the lamella pre-tilt?

- While the text is in general well readable, there are several instances of highly repetitive sentence structures, missing articles, and similar *minor* “faux pas”, which should be corrected.

- “Bold” claims. There are several claims/statements that might not be true as generally as they are presented. E.g.

- “... reconstruct the protein complex at near atomic resolution free of model bias.”

With less bias, but free? How was this assessed?

- “Cryo-electron tomography (cryo-ET) was developed to study protein structure in situ” Are the authors sure about that?

- “the throughput of cryo-ET is still only approximately 1/30 that of single particle data collection”

Where is this from?

- “However, how to obtain high-resolution structures remains uninvestigated.”

Arguably other groups have already demonstrated high-resolution TM for STA-like reconstructions.

Etc.

Reviewer #3 (Remarks to the Author):

The authors present work and software that extends their previous work and software by including GPU support and processing in-situ cryo-FIB data. I have experience with cryo-FIB/SEM, cryo-ET, and cryo-EM, but little biology knowledge, so I will focus on the technical and practical aspects of this work. Overall, I think this manuscript and software are beneficial for the field, particularly because this is the first attempt at high-resolution SPA in FIB-milled cells that I know of. I think this work should be published after concerns are addressed, which are only minor and moderate concerns. I list my questions and comments below ('-' are minor comments/questions and '+-' are moderate comments/questions):

-Lines 21-23: Multiple labs are starting to produce cryo-ET averages from cryo-FIB-milled lamellae in the 4-5 angstrom resolution range with the new slew of software that can correct for much of the alignment errors present in tilt-series (Warp/M, Relion4, EMAN2, EMClarity), so I'm not sure if this is a big motivating factor for doing SPA on lamellae. I think the real motivating factor is throughput - SPA collection is 10-100 times faster than cryo-ET collection. I would emphasize this.

-Line 35: SPA wasn't developed to reach near-atomic resolution - that took decades of work. Also, citations 1 & 2 arguably show atomic resolution, not near-atomic resolution.

-Line 40: Cryo-ET was developed not only to look at protein structure in-situ, but also other cell parts like membranes and ultrastructure. Importantly also is that cryo-ET gives contextual information.

-Line 44: I think the software by the Scheres lab and Kudryashev lab should also be added to this list of references: doi.org/10.1038/s41467-020-17466-0 doi.org/10.1101/2022.02.28.482229

-Line 51: The Waffle Method paper (doi.org/10.1038/s41467-022-29501-3) should be added because it reports on a new milling procedure for reducing lamellae stress.

-Line 63-65: Warp/M's multi-species refinement allows for higher-resolution alignment of less abundant proteins by co-refining proteins with higher abundance (e.g. ribosomes). This is, however, an option limited to specimens that have high abundance, conformationally homogeneous proteins together with the protein of interest.

-References 7 and 16 are the same.

-Reference 27 shows Egelman as the first author, but he's not.

-Reference 30 should be updated: doi.org/10.1126/science.abn1934

-This recent manuscript showing the 80S yeast ribosome at 4.5 angstroms by cryoET should be referenced: doi.org/10.1101/2022.06.16.496417

-Line 83 & paragraph 243: The correct reference is Lucas et al., not Bronwyn et al.

-Line 119: How is 'n' experimentally determined? Or is this a heuristic parameter?

-Line 316: What percentage of particles included overlapping densities from other biomolecules?

-Line 331: Where did the number 'dozens' come from for 'small' proteins?

-Line 362: I just want to confirm: No energy filter was used? Line 428 makes it seem like you're using an energy filter to estimate ice thickness...

-Line 366: 'model' >> 'mode'

-Can particle z-heights be estimated reliably, like is done in Lucas et al.? Perhaps with CTF refinement?

+I could not find a description of how the ab-initio model was created for Relion single particle alignment. I am worried that the authors used the high-resolution template as a starting model.

+It would be nice to see histograms of GisSPA's picking accuracy; ie. After SPA refinement, show a histogram of displacements of the final x,y particle positions relative to the initial x,y positions, and a second histogram of the angular displacements of the final particle in-plane euler angles relative to the initial euler angles.

-The cryoEM field always benefits when authors upload their raw data to EMPIAR. Consider doing so for the data generated in this manuscript; the field will thank you=)

I hope you're all well.

Best wishes,

Alex Noble

Reviewer #1 (Remarks to the Author):

The manuscript from Cheng et al describes a new package for 2D template matching. The novelty, in this case, is lacking as the method was described already. Nevertheless, the GPU acceleration is worth attention as the previous implementations of this method requires multiple days to process a single micrograph.

The article is well written and easy to understand.

In order to better follow whether the work is worth the novelty, I would like to see a proper comparison with the existing methods (e.g. process the same data using CISTem and compare the difference in accuracy). Further, by extracting the particles with no randomisation of the angles, the template bias will be inevitable, and the resolution reported may be effectively an over-estimation.

In principle, the experiment is easy to perform.

To properly review the article, I downloaded the software, which runs with minor bugs at this stage, and I have noticed a difference in performance compared to CISTem, making it faster, but data displaying more charging is also less accurate.

An in depth analysis of the performance, accuracy and bias will definitely make the article worth publication.

Thank you for the comments and raising important points helping to improve our work. We would like to explain the reference bias problem first. In isSPA, the signal of 3D references that was used to “locate” target protein in micrographs is usually limited to 8 Å. In such a case, any signals with frequency higher than $1/8 \text{ \AA}^{-1}$ are not included in the calculation of projection match. To test the template bias problem in isSPA, we had used a test dataset-HSV (an 120nm-diameter icosahedral virus) dataset, which had been analyzed by single particle method previously, so we knew the ground truth (the location and orientation of capsid proteins) on each virus in micrographs. We used GisSPA to “localize” the capsid proteins on the virus and the top solutions beyond the threshold was so called localized particles. Taking the ground truth as positive controls, we were able to distinguish between true particles with translational and rotational parameters close to the ground truth and false particles in the localized particles dataset. We performed local refinements as we do in isSPA on these localized particles including false particles and true particles. The FSC of the reconstruction showed a sudden drop at $1/8 \text{ \AA}^{-1}$ as marked by an arrow in Fig1a. False particles have the reference bias problem. We selected the known false particles in the refined dataset and made two reconstructions using two half datasets of false particles. We calculated the FSC curve of false particles as shown below. The FSC value at the frequency lower than $1/8 \text{ \AA}^{-1}$ is close to 1, which indicates strong reference bias problem. However, the FSC drops quickly to 0 at the frequency beyond $1/8 \text{ \AA}^{-1}$ (Fig1b). Thus, the particle “localization” step and the local refinement do not create bias beyond $1/8 \text{ \AA}^{-1}$. The possible reason is that we

performed only local refinements with an initial step size of 1.8 degrees, the false particles are unlikely to be able to fit right with the high-resolution signals above $1/8 \text{ \AA}^{-1}$ in such a search with highly restrained parameters. The true particles were selected and the FSC of true particles was shown in Fig1b. We performed a new round of local refinements on true particle and the reported resolution based on 0.143 FSC further improved of $\sim 0.3 \text{ \AA}$. Therefore, in our case, false particles caused reference bias problem at frequency lower than $1/8 \text{ \AA}^{-1}$. At frequency beyond $1/8 \text{ \AA}^{-1}$, the including of false particles disturbs the alignment and limits the resolution.

Figure 1. (a) FSC curve of the mixture of true particles and false particles. (b) FSC curves of reconstructions from true particles (green) and from false particles (red).

For a real *in situ* dataset, we don't have positive controls. As we shown in the previous paper¹, the sorting method can efficiently separate the false particles from true particles (please also see our response to reviewer #2). For different datasets processed by isSPA, we found similar phenomenon as described above that the removing of false particles helps to improve the resolution. In this dataset from cellular lamella, we tried to localize 1.5MD PSII complexes using the frequencies up to $1/9 \text{ \AA}^{-1}$. The FSC of false particles given by sorting method as shown in Fig2 dropped sharply to 0, which is similar to that in HSV dataset. A new round of local refinements of remaining particles increased the resolution by 1 \AA to 4.7 \AA . In the revised manuscript, we have added 377 micrographs, 33,660 particles were selected via 3D classification. Thus, we included these particles and applied sorting method to the joined particle dataset. The resolution of the final reconstruction reached 3.9 \AA . We present the densities of the map before sorting and of the final map as Fig3.

Figure 2. FSC curves of reconstructions from removed particles (red) and reserved particles (green) distinguished by sorting.

Figure 3. Densities of map at 5.7 Å before sorting (left, gray) and densities of the final map at 3.9 Å (right, green).

In addition, during the local refinements of remaining particles, the initial model we provided to RELION was low-pass filtered to 8–10 Å. Without a high-resolution model having side chain densities, the chance that reference biased map have a high-resolution feature that resembles correct side chain densities is very low.

Moreover, when we only use part of a complex as the reference, other densities not included in the reference but occurred in the reconstructed map can be another criterion to determine whether the map is dominated by reference bias. The reference bias effect only produces features of the reference itself, so there is no chance that we can recover the densities which are not included in reference. With similar occupancy, the density quality of these two parts should be close. In our calculation, when we used PBS as the reference to localize particles, we also reconstructed the PSII complex connected to the PBS and the map quality of PSII complex was close to PBS region. When we calculate 70S ribosome in this manuscript, we used the 50S part as the reference, and the result of 3D classification showed that only some of 50S particles are connected to 30S, the ratio of 30S to 50S is 0.83, which is close to the tomograph study result (0.7)². The resolution of 50S and 30s was 6.73 Å and 8.93 Å, respectively. In the results of cisTEM³ from the same dataset, the resolution of 50S and 30s was 4.3 Å and 15 Å, respectively,

as stated in their paper. It indicates that there might be strong reference bias problem in their reconstruction. Reviewer #2 of their paper raised the same comment to their work.

In conclusion, the three points above show that our approach does not introduce bias at high frequencies beyond the frequency threshold used in particle localization.

Next, we will compare the difference between cisTEM and GisSPA in 3 main points.

(1) Particle localization.

GisSPA prefers to use cryo-EM map as a 3D template when it is available, but cisTEM uses the simulated map generated by PDB file as a 3D template. GisSPA uses the derived weighting function by maximizing the SNR of CC, the weight at different frequencies can be adjusted depends on the data. cisTEM uses a whitening filter calculated from the micrograph.

(2) Exclude false particles.

In the step of “localize” particles, we found that it is hard to use a threshold to only localize true particles. Thus, GisSPA removes false particles through non-alignment 3D classification and sorting method. cisTEM used a fitted threshold to “localize” particles, they didn’t report a method to remove false particles.

(3) Particle refinement.

GisSPA only performs local refinement with small step size on the particles to improve the resolution. cisTEM doesn’t perform refinement. It reconstructs map using parameters obtained from 2D matching as was described in their paper³.

We take the available *M. pneumoniae* dataset as the test data for the overall comparison. The result of the overall processing of the two approaches is as described above. The reported resolutions of 50S (6.7 Å) and 30S by GisSPA (8.9 Å) are close to each other. However, the resolutions of 50S (4.3 Å) and 30S (15 Å) by cisTEM differ greatly as they described, indicating strong reference bias effect in their reconstruction. This effect was also stated by themselves in their paper.

Next, we directly compare the localization step. We downloaded the 3D template of 50S “2DTM_template.mrc” (created from a PDB file) used in 2DTM in cisTEM from EMPIAR, using the same parameter as doi: [10.7554/eLife.68946](https://doi.org/10.7554/eLife.68946) except for the defocus value search. The resolution cut was set to 8 Å. However, this value was not mentioned in their paper. Since the defocus error of 50nm barely affect CTF at the frequency lower than $1/8 \text{ \AA}^{-1}$, we skipped the defocus value search step which is extremely time consuming (2h for a micrograph without defocus search and ~ 1 day for a micrograph with defocus search). 5,000 “localized” particles were picked according to the threshold 9.0. To further analysis the data, we processed these particles followed the workflow of GisSPA. 2,824 of them were classified as the good class by a round of non-alignment 3D classification as implemented in isSPA. But the result of local refinement on the 2,824 particles showed some artifacts annotated by a yellow rectangular box in a 2D slice of the 3D map below:

Figure 4. A central 2D slice of the 3D map reconstructed from particles picked by cisTEM when using a 3D model generated by PDB. The artifacts were indicated by a yellow rectangle box in the right panel.

So, we used the EM map mentioned in our manuscript instead of the map created based on the PDB model as the 3D model and searched by cisTEM again, 4,861 localized particles were picked by the suggested threshold of 7.5. By applying a round of non-alignment 3D classification, 3,225 particles were classified into the good class. The portion of true particles is ~ 0.66 . For comparison, 5,554 particles were localized by GisSPA and 4,560 particles were classified into the good class. The portion of true particles is ~ 0.82 .

However, we can't reproduce the high-resolution results (50s at resolution of 4.3 Å) from *doi: 10.7554/eLife.68946* and we cannot figure out how the model bias occurred in *doi: 10.7554/eLife.68946* produced, since they didn't provide the cutoff value of frequency that was used for localization and the detailed parameters for the reconstruction.

In addition, we took the cellular lamellae as another dataset for comparison. We used the 8 Å resolution structure of 2.3MD protein, which was segmented from PBS, as the reference. In order to calculate the accuracy, we localized the 2.3-MD protein on 100 micrographs using 3-degree out-of-plane angular step and 2-degree in-plane angular search, and the high-resolution cutoff was 8 Å. 28,388 particles were selected under the threshold 6.5. The same number of particles were picked with GisSPA using the same search parameters for comparison. We performed a non-alignment 3D classification of the particles located by both approaches. Since the reference contains only the part of the 2.3MD protein in the PBS, so the false particles only have the densities of the 2.3 MD protein. We counted the particles that exhibit the entire PBS feature as the true particles. The result of 3D classification shows that the percentage of true particles picked by GisSPA is 29.7%, and the percentage of true particles picked by cisTEM is 19.8%. The two methods may differ more on smaller proteins, but when we localize a smaller protein, the proportion of true particles is too low and the result from 3D classification was not accurate enough. So, the test on smaller protein is not performed.

In conclusion, our current data shows that GisSPA is more accurate than Cistem. The other major reason causes the difference in the final results is that isSPA designed strategies to exclude false particles during refinement.

"To properly review the article, I downloaded the software, which runs with minor bugs at this stage, and I have noticed a difference in performance compared to CISTEM, making it faster, but data displaying more charging is also less accurate."

We are sorry for the bugs and we will continue to improve the program to make it easier for users.

There are two reasons why GisSPA is faster, one is that we split the micrographs as windows, and the other is that we skipped the CCG normalization⁴, which is very time-consuming.

We're sorry we don't quite understand the features of data displaying more charging, we guess that you mean data with large drifts caused by charging or data with ice contamination. So far, we have not processed with data with abnormal motion values. It is possible that the reason for the less accuracy in the charging data is that the step of normalization was lost in GisSPA. We have added the normalization operation based on ref.4 as a new option. As to whether normalization works well on special data, we need to do more test. Could you please tell us is there any public data with charging? Thus, we can download it for the test in future.

Reviewer #2 (Remarks to the Author):

Review of Zhang et al. 2022, "Determining protein structures in cellular lamella at pseudo-atomic resolution by GisSPA"

In the manuscript, Zhang et al. present their GPU-accelerated software - GisSPA - for 2D template matching on cellular cryo-FIB lamellas. The authors present three examples of sub-nanometer structures – Phycobilisomes (*P. purpureum*, C2 symmetry, 3.4 Å), photosystem II complex (4.7 Å, C4), and ribosomes (6.9-8.9 Å, *M. pneumoniae*, C1) – reconstructed from their hits and use a special sorting algorithm to reduce the template bias introduced by searching their projection images against high resolution reference structures.

We apologize for not describing clearly in the manuscript that the photosystem II complex was reconstructed with C2 symmetry instead of C4, and in the revised manuscript we have improved it to 3.9 Å resolution by adding 377 more micrographs.

In summary, the paper presents interesting results and an alternative to subtomogram averaging (STA) for targets that have indeed been more difficult to resolve at high resolution by STA. While the method for sure could be interesting for some, overall it is questionable that novelty, applicability, and efficiency make GisSPA and its manuscript suitable for Nature Communications.

We thank the reviewer for the time and efforts in reading and assessing our manuscript. We have made the point-to-point responses below.

Major:

- Novelty

As far as this reviewer can tell, nothing major has changed from isSPA, except for the GPU-acceleration. Admittedly, this is a major improvement and for sure a lot of work has gone into that. While this has reduced the time needed for performing the template search from months (60 days for 200 CPUs, line 124) to days, there is still a significant commitment of resources necessary to perform the template matching task.

Additionally, a similar approach has already been introduced by Grigorieff et al.

We thank the reviewer for the comment. It is a concern in the field that whether near-atomic resolution (<4Å) can be achieved on cell lamellas to enable the atomic modeling, and it has not yet been achieved by now. This work proved that the near-atomic resolution structure can be achieved from the cellular lamellas by GisSPA. Our data demonstrated that GisSPA is a practical tool that can be used to achieve near-atomic resolution reconstruction on cellular lamella relatively easily with appropriate molecular weight.

Our previous manuscript about isSPA has been preprinted on bioRxiv on Oct. 19th, 2020. However, there is no applications on cellular lamellae in the previous work. Grigorieff et al. has published two works based on *M. pneumoniae* cells (published on Jun. 6th, 2021,

submitted on Mar. 30th, 2021) and *S. cerevisiae* lamella (published on Aug 25, 2022), respectively. However, their work showed severe template bias effect as stated by themselves and also mentioned by the reviewers. We have discussed about it above (see response to Reviewer #1). They need to solve this bias problem to make 2DTM in cisTEM practical. Therefore, high-resolution in-situ structure determination on cryo-lamellae using 2D template matching has not yet been established before this work.

We used 3 days to process 1500 micrographs on 7 GTX1080 GPUs for “localization” of target protein. The same number of A100 GPUs will cut the running time to less than 1 day. The processing time of localization step by GisSPA is relatively short compared with the time taken to resolve the *in-situ* structure to near-atomic resolution by tomography and STA.

- Efficiency

As mentioned before, this reviewer recognizes the improvement due to GPU acceleration, but running the software on seven GTX 1080 GPUs over three days is still a major time/resource investment hardly compatible with “screening” different targets and “standard” single-particle like workflows. Overall, this will limit the applicability of the GisSPA to specific projects with clear targets.

We agree with the reviewer that the GisSPA takes more time than the single particle approach. For specific projects with clear targets, such as membrane proteins embedded in liposomes⁵ and spike proteins visually protruding to the outside of viruses in cryo-EM images⁶, single-particle like workflows can pick certain particles and solve the structure.

For the specific projects with clear targets, GisSPA does take more time than single-particle like workflows, but GisSPA has some advantages in other aspects. Yao et.al. has published their structure of AcrB determined from liposomes, in the paper they used 65,317 particles from 5,757 micrographs to reconstruct the final map at 3.9 Å by single-particle like workflows. However, they only solved the protein inward, the outward remained unsolved, so they collaborated with us to solve the protein outward (unpublished data). The result of isSPA showed a 3.9 Å resolution map of protein inward (21,912 particles) and a 4.2 Å resolution map of protein outward (5,401 particles) as Fig6. It is worthy to notice that the results were calculated from only 700 micrographs, far less than the number of micrographs used in single-particle like workflows.

Figure 5. Cryo-EM maps of AcrB embedded inside (purple) and outside (pink) liposome, respectively. Alpha helices (blue) were segmented to present the density quality.

In addition, the major aim of developing GisSPA is to solve *in situ* structures in cellular environment. The proteins in cellular environment usually can't be picked by single particle like methods.

- Constraints

- Lamella thickness

The authors rightfully point out that GisSPA requires relatively thin lamellas (<170 nm). Thickness of lamellas is however inversely proportional to their stability/quality. Producing such lamellas is more difficult and not as easily automated, limiting the sample preparation for GisSPA.

We thank the reviewer for the comment. Our precision recalls (Fig.3 in manuscript) for proteins with different molecular weights indicates that only small proteins with molecular weight close to 1.1MD or fewer require thin lamellas < 170 nm. For some larger complexes, a greater range of thicknesses are acceptable.

The current FIB workflow can easily achieve lamellas around 200nm^{7,8}. Our FIB data indicates that the half of the lamellas are thinner than 170 nm with a careful FIB thinning.

- Molecule weight

The authors suggest that the lower limit for detection is ~ 1.1 MDa. Of course, there is a limit for template matching and STA, too, however significantly smaller complexes such as Rubisco (~0.5 MDa) have already been solved in situ (Engel B. et al.). This furthermore limits the general applicability of GisSPA.

We thank the reviewer for the comment. The goal of GisSPA is to achieve near-atomic resolution with the help of high throughput data collection. Our data demonstrated that we have the potential to effectively localize the protein and resolve it to near-atomic resolution when the molecular weight is larger than 1.1 MD. Smaller proteins such as Rubisco in cellular lamella can be picked from tomograms. However, till now, the resolution of Rubisco in cellular lamella by STA is 16 Å. This resolution is sufficient for answering some biological questions, but such a resolution is not the goal of GisSPA.

In addition, 1.1 MD is the limit on cellular lamellae. In other systems, such as membrane proteins embedded in liposome, we resolved the 350 kD protein to 3.9 Å as described above, and in the purified Carboxysomes, we resolved the 500 kD Rubisco to 3.7 Å. Therefore, the molecular weight limit of GisSPA on non-sectioning sample is fewer than 1.1 MD.

- Model bias and claim of "bias free"

The model bias is mitigated by a “sorting algorithm” implemented already in isSPA, where “particles (···) are sorted according to their scores” (isSPA publication in The Innovation 2, 100166, November 28, 2021), which - unless this reviewer fails to see a more important step - seems like an oversimplification and stretch of the word “algorithm”. Even a bimodal distribution is not a sufficient criterion for selecting true and false positives since CCCs between template matching runs are not comparable (even when normalized). This is a misconception sometimes also presented in STA papers.

We thank the reviewer for raising this problem. We are sorry for the inappropriate use of word “algorithm”, maybe sorting method is more suitable. We have changed “sorting algorithm” to “sorting method” in the revised manuscript.

In our previous isSPA publication, we used HSV-2 dataset as the test data. We have the single particle result of HSV-2. Thus, we knew the position and orientation of each capsid protein. We used these capsid proteins as positive controls. After we “localized” capsid proteins on the HSV2 by isSPA, we differentiated false particles from true particles with the help of the positive controls. After the local refinement of the dataset containing both false particles and true particles, we calculated phase residuals of true particles and false particles, respectively. The phase residuals and the corresponding number of particles are showed as below. In addition , number of false particle (scattered points in red) and true particle (scattered points in green) at each phase residual range are plotted. It is worthy to point out that the red dot and green dot are not a result from a bimodal distribution fitting. Then we tried to fit the overall data using the bimodal distribution. The results showed that the fitted results (solid lines) matched well with the real data distribution (scattered points). When we removed large portion of false particle, the remaining particles were fewer, but the resolution was improved instead (from 4.3 Å to 4.0 Å).

Figure 6. Number of particles relative to phase residual of true particles (green scatters), false particles (red scatters) and total particles (black solid line). Bimodal fitting of the black solid line (red solid line and green solid line).

In the real data, we don't have positive controls. But the distribution is similar and can be fit with bimodal distribution. we throw away most of the particles located in the left Gaussian peak in the same way, and the final resolution is effectively improved. In two different datasets, the resolution of Rubisco improved from 4.3 Å to 3.9 Å after applying sorting method, and the resolution of PSII complex improved from 5.7 Å to 4.7 Å (further improved to 3.9 Å after adding 377 micrographs and applying a second round of sorting) in this manuscript.

Minor:

- "therefore, rotating at the stage before data collection is operated to make sure the normal direction is along the direction of the incident electron beam and large protein motion is avoided."

Do the authors mean that they account for the lamella pre-tilt?

We thank the reviewer for pointing this out. We rotated the stage to compensate for the lamella pre-tilt. We have clarified this in the revised manuscript.

- While the text is in general well readable, there are several instances of highly repetitive sentence structures, missing articles, and similar *minor* "faux pas", which should be corrected.

Thank you for pointing out this, we have sent our manuscript for further polishing and made corrections in the revised manuscripts.

- "Bold" claims. There are several claims/statements that might not be true as generally as they are presented. E.g.

- "... reconstruct the protein complex at near atomic resolution free of model bias."

With less bias, but free? How was this assessed?

Our data (please see our response to Reviewer #1) shows that FSC values of high-resolution frequencies ($>> 8 \text{ \AA}$) are unlikely to be biased by false particles. Therefore, we proposed model bias free, however, the remaining false particles cause little bias within 8 Å, so we delete this description in the revised manuscript.

- "Cryo-electron tomography (cryo-ET) was developed to study protein structure in situ" Are the authors sure about that?

We apologize for the inappropriate description, and we have modified it in the revised manuscript as below:

"Cryo-electron tomography (cryo-ET) is the most popular tool to study protein structure in situ"

- "the throughput of cryo-ET is still only approximately 1/30 that of single particle data collection"

Where is this from?

This is an estimation according to the throughput of beam-image-shift tomography data collection⁹ (288 tilts per 24h, K2) and that of beam-image-shift single particle data collection (~8,000 micrographs per 24 hours, K2) currently.

- "However, how to obtain high-resolution structures remains uninvestigated."

Arguably other groups have already demonstrated high-resolution TM for STA-like reconstructions.

We are sorry for the confusing. Here we would like to state that the work by *Rickgauer et al.* did not investigate how to obtain high-resolution structures in overlapping densities, so we developed isSPA to build a complete workflow. To avoid misunderstanding, we changed the description as below:

"Rickgauer et al. used a high-resolution reference and a whitening filter to localize target proteins within overlapping densities. However, in this work they didn't provide a complete workflow to obtain a high-resolution structure from localized particles with overlapping densities. We then developed isSPA method."

Reviewer #3 (Remarks to the Author):

The authors present work and software that extends their previous work and software by including GPU support and processing in-situ cryo-FIB data. I have experience with cryo-FIB/SEM, cryo-ET, and cryo-EM, but little biology knowledge, so I will focus on the technical and practical aspects of this work. Overall, I think this manuscript and software are beneficial for the field, particularly because this is the first attempt at high-resolution SPA in FIB-milled cells that I know of. I think this work should be published after concerns are addressed, which are only minor and moderate concerns. I list my questions and comments below ('-' are minor comments/questions and '+-' are moderate comments/questions):

We thank the reviewer for his recognition of our work and we have made our responses point-to-point as below.

-Lines 21-23: Multiple labs are starting to produce cryo-ET averages from cryo-FIB-milled lamellae in the 4-5 angstrom resolution range with the new slew of software that can correct for much of the alignment errors present in tilt-series (Warp/M, Relion4, EMAN2, EMClarity), so I'm not sure if this is a big motivating factor for doing SPA on lamellae. I think the real motivating factor is throughput - SPA collection is 10-100 times faster than cryo-ET collection. I would emphasize this.

We agree with the reviewer and we thank the reviewer for the comments and helpful suggestions. We have made changes in the revised manuscript:

***“Cryo-electron tomography is a major tool used to study the structure of protein complexes in situ. However, the throughput of tilt-series image data collection is still quite low. Here, we show that GisSPA, a GPU accelerated program, can translationally and rotationally localize the target protein complex in cellular lamellae, as prepared with a focused ion beam, using single cryo-electron microscopy images without tilt-series, and reconstruct the protein complex at near-atomic resolution. GisSPA allows high-throughput data collection without the acquisition of tilt series images and reconstruction of the tomogram, which is essential for high-resolution reconstruction of asymmetric or low-symmetry protein complexes. We demonstrate the power of GisSPA with 3.4-Å and 3.9-Å resolutions of resolving phycobilisome and tetrameric photosystem II complex structures in cellular lamellae, respectively. We present GisSPA as a practical tool that facilitates high-resolution in situ protein structure determination.*”**

-Line 35: SPA wasn't developed to reach near-atomic resolution - that took decades of work. Also, citations 1 & 2 arguably show atomic resolution, not near-atomic resolution.

We are sorry for the wrong descriptions and we have made corrections in the revised manuscript as below:

“After decades of development, single-particle cryo-electron microscopy (cryo-EM) has determined the structure of proteins in solution to near atomic resolution or even to atomic resolution on apo-ferritin.”

-Line 40: Cryo-ET was developed not only to look at protein structure in-situ, but also other

cell parts like membranes and ultrastructure. Importantly also is that cryo-ET gives contextual information.

**Thank you for pointing this out, we have made corrections in the revised manuscript:
“Cryo-electron tomography (cryo-ET) is the most popular tool to study protein structure in situ”**

-Line 44: I think the software by the Scheres lab and Kudryashev lab should also be added to this list of references: doi.org/10.1038/s41467-020-17466-0
doi.org/10.1101/2022.02.28.482229

We are sorry for missing the two important references. We have added the two references to the revised manuscript.

-Line 51: The Waffle Method paper (doi.org/10.1038/s41467-022-29501-3) should be added because it reports on a new milling procedure for reducing lamellae stress.

We are sorry for missing this citation, and we have added this reference to the revised manuscript.

-Line 63-65: Warp/M's multi-species refinement allows for higher-resolution alignment of less abundant proteins by co-refining proteins with higher abundance (e.g. ribosomes). This is, however, an option limited to specimens that have high abundance, conformationally homogeneous proteins together with the protein of interest.

Thank you for the suggestions. We have made changes in the revised manuscript as you suggested.

-References 7 and 16 are the same.

Thank you for pointing out this, and we have corrected this in the revised manuscript.

-Reference 27 shows Egelman as the first author, but he's not.

Thank you for pointing out this, we are sorry for the mistakes. This mistake may be caused by incorrectly citing editor's name as first author by “Mendeley” software and we have corrected this in the revised manuscript.

-Reference 30 should be updated: doi.org/10.1126/science.abn1934

Thank you for point out this, we have updated the newest version of this citation in the revised manuscript.

-This recent manuscript showing the 80S yeast ribosome at 4.5 angstroms by cryoET should be referenced: doi.org/10.1101/2022.06.16.496417

Thank you and we have added this reference in the revised manuscript.

-Line 83 & paragraph 243: The correct reference is Lucas et al., not Bronwyn et al.

We are sorry for the mistake. We have made changes in the revised manuscript.

-Line 119: How is 'n' experimentally determined? Or is this a heuristic parameter?

“n” is a heuristic parameter. We have tested different n in our previous work (The Innovation, 2021), the results showed that when n increased from 0 to 3, the improvement of precision-recall is considerable, but when n increased from 3 to 6, the precision-recalls almost remained unchanged. We also tested n value from 3 to 10 using this dataset, the result was consistent with the previous results.

-Line 316: What percentage of particles included overlapping densities from other biomolecules?

For icosahedral virus, we used block-based reconstruction. On these viruses (PBCV-1, ASFV, alphaviruses, HSV), almost all blocks are overlapped with other densities, but the degree of overlapping depends on the sample. For virus sample, blocks distributed around the edge of shell has fewer overlapping densities than in the middle areas.

-Line 331: Where did the number ‘dozens’ come from for ‘small’ proteins?

We are sorry for the unsuitable expression of ‘dozens’, actually the differences of required data are related to the target resolution. Higher resolution causes larger differences, so this part was modified to:

“The B factor values of PBS in cellular lamella and purified proteins are close, probably because the PBS is large so that the decrease of the SNR from factors such as sample thickness barely affect the alignment. The accuracy of alignment may get reduced when the target protein becomes small, indicating that smaller proteins have larger B-factor values than large proteins.”

-Line 362: I just want to confirm: No energy filter was used? Line 428 makes it seem like you're using an energy filter to estimate ice thickness...

The energy filter was used in our data collection and we have added this part in the revised manuscript.

“The lamellas of P. purpureum data were collected on a FEI Titan Krios EM operated at 300 kV equipped with a Gatan K3 Summit detector and an energy filter (20 eV slit width).”

-Line 366: ‘model’ >> ‘mode”

We are sorry for the spelling error and we have corrected it in the revised manuscript.

-Can particle z-heights be estimated reliably, like is done in Lucas et al.? Perhaps with CTF refinement?

Particle z-heights can't be estimated reliably through particle “localization” step, this is because that only frequencies below $1/8 \text{ \AA}^{-1}$ are included in calculating, and signals at this frequency range is not enough sensitive to defocus variations. If Lucas et al. search for the per-particle defocus value, they must use a template with a resolution up to $\sim 4 \text{ \AA}$. In that situation, it will be difficult to remove false particles, thus causing template bias problem. Therefore, we choose to correct the defocus values by CTF refinement implemented RELION or cryoSPARC. Even though, there is no guarantee that the defocus value of every particle is accurately corrected. In our opinion, tomogram reconstruction is the most reliable way for accurate z heights determination.

+I could not find a description of how the ab-initio model was created for Relion single particle alignment. I am worried that the authors used the high-resolution template as a starting model.

We are sorry for missing the description about the ab-initio model. Before auto-refinement, at least one round of non-alignment 3D classification was performed (using low-pass filtered 3D template as initial model). Then the 3D reconstruction generated by particles detected by GisSPA in the best class was used as the initial model in auto local refinement and the lowpass filter was set to 20 Å. When we refine particles after sorting, the low-pass filter is generally set to 8-10 Å. We have added this part of description to our revised manuscript.

+It would be nice to see histograms of GisSPA's picking accuracy; ie. After SPA refinement, show a histogram of displacements of the final x,y particle positions relative to the initial x,y positions, and a second histogram of the angular displacements of the final particle in-plane euler angles relative to the initial euler angles.

Thank you for your helpful suggestions, we have calculated the histograms of differences of x,y and euler values before and after SPA refinement of the PBS particles, and the results are listed below:

The units of the horizontal coordinates are angstroms and degrees, respectively, and the

vertical axis indicates the number of PBS particles. As we can see, the mean x,y shifts is around 1.2 angstroms and the orientational shifts is resented as a Gaussian-like distribution with a mean of ~1.36 degrees.

-The cryoEM field always benefits when authors upload their raw data to EMPIAR. Consider doing so for the data generated in this manuscript; the field will thank you=)

We thank the reviewer for the suggestion. The data in this manuscript is collected by our collaborators and it is also the source data for a company manuscript (2022-06-10034A), which is under review now. It is hard for us to make the data public at this time. Our collaborators have collected some tilt-series on the same sample to perform STA (~48,000 C1 particles @ 5.6 Å using emclarity), and they calculated the Rosenthal-Henderson plot of PBS particles, so far, the B-factor was estimated as 282.5 Å² and the intercept was -0.04634. Therefore 4,746,319 PBS particles were needed for a 4 Å resolution structure determination by STA on this sample, about 150 times as much as GisSPA approach (31,862 particles, B-factor 131.1 Å²). We think that sharing the two sets of data (tilt series and single micrographs) for more analysis and will be beneficial for the comparison between the two approaches. We would like to upload the two datasets with the agreement of our collaborators after the publish of both papers.

I hope you're all well.

Best wishes,

Alex Noble

1. Cheng, J., Li, B., Si, L. & Zhang, X. Determining structures in a native environment using single-particle cryoelectron microscopy images. *The Innovation* **2**, (2021).
2. O, F. J. *et al.* *In-cell architecture of an actively transcribing-translating expressome*. *Science* vol. 369 <https://www.science.org> (2020).
3. Lucas, B. A. *et al.* Locating macromolecular assemblies in cells by 2d template matching with cistem. *Elife* **10**, (2021).
4. Saxton, W. O. *MOTIF DETECTION IN QUANTUM NOISE-LIMITED ELECTRON MICROGRAPHS BY CROSS-CORRELATION* *. *Ultramicroscopy* vol. 2 (1977).
5. Yao, X., Fan, X. & Yan, N. Cryo-EM analysis of a membrane protein embedded in the liposome. doi:10.1073/pnas.2009385117/-/DCSupplemental.
6. Ke, Z. *et al.* Structures and distributions of SARS-CoV-2 spike proteins on intact virions. *Nature* **588**, 498–502 (2020).
7. Kelley, K. *et al.* Waffle Method: A general and flexible approach for improving throughput in FIB-milling. *Nat Commun* **13**, 1857 (2022).
8. Berger, C. *et al.* Structure of the Yersinia injectisome in intracellular host cell phagosomes revealed by cryo FIB electron tomography. *J Struct Biol* **213**, 107701 (2021).
9. Bouvette, J. *et al.* Beam image-shift accelerated data acquisition for near-atomic resolution single-particle cryo-electron tomography. *Nat Commun* **12**, 1957 (2021).

REVIEWERS' COMMENTS

Reviewer #1 (Remarks to the Author):

I am happy with the response.

I unfortunately am not aware of public datasets with more or less charging, but I do have some of my own, and I am happy to share those to improve the software.

Best wishes

Alex de Marco

Reviewer #2 (Remarks to the Author):

In their revised manuscript, Cheng et al have addressed the concerns not only raised by me, but also by the other reviewers.

While I still do not fully agree with the claim of “bias free”,

a) because bimodal distribution only considers low-scoring false positives, not high scoring ones.

b) are $\pm 0.3/1 \text{ \AA}$ really significant at $8/9 \text{ \AA}$ global resolution?

the approach put forward in this paper is a “realistic solution”, i.e. doing what can be done in the absence of a ground-truth calibration.

More of a comment: The claim that “It is a concern in the field that whether near-atomic resolution ($<4 \text{ \AA}$) can be achieved on cell lamellas to enable the atomic modeling, and it has not yet been achieved by now” is just not true. See for example Tegunov et al. Nat Met 2021. (and others).

In general, it may be more useful to focus on the achievements of GisSPA and not what others (allegedly) did not achieve.

Lastly, some rebuttal figures (e.g. 6&7) would actually be useful as supplementary figures. They should be included in the final version.

Looking forward to testing GisSPA in cellular lamellas!

Reviewer #3 (Remarks to the Author):

I thank the authors for addressing reviewers' comments. I recommend the manuscript for publication.

Reviewer #1 (Remarks to the Author):

I am happy with the response.

I unfortunately am not aware of public datasets with more or less charging, but I do have some of my own, and I am happy to share those to improve the software.

Best wishes

Alex de Marco

Thank you very much and we look forward to contacting you!

Reviewer #2 (Remarks to the Author):

In their revised manuscript, Cheng et al have addressed the concerns not only raised by me, but also by the other reviewers.

While I still do not fully agree with the claim of “bias free”,

a) because bimodal distribution only considers low-scoring false positives, not high scoring ones.

b) are $\pm 0.3/1 \text{ \AA}$ really significant at $8/9 \text{ \AA}$ global resolution?

the approach put forward in this paper is a “realistic solution”, i.e. doing what can be done in the absence of a ground-truth calibration.

More of a comment: The claim that “It is a concern in the field that whether near-atomic resolution ($<4\text{\AA}$) can be achieved on cell lamellas to enable the atomic modeling, and it has not yet been achieved by now” is just not true. See for example Tegunov et al. Nat Met 2021. (and others).

In general, it may be more useful to focus on the achievements of GisSPA and not what others (allegedly) did not achieve.

Lastly, some rebuttal figures (e.g. 6&7) would actually be useful as supplementary figures. They should be included in the final version.

Looking forward to testing GisSPA in cellular lamellas!

We thank the reviewer for the comments and helpful suggestions. We have made our responses below.

a) because bimodal distribution only considers low-scoring false positives, not high scoring ones.

We agree with the reviewer that only low-scoring false positives are removed from the particle set, so we have already deleted the claim of “bias free” in the previously revised manuscript.

b) are $\pm 0.3/1 \text{ \AA}$ really significant at $8/9 \text{ \AA}$ global resolution?

We are sorry for the confusing, the resolution improvement of $\pm 0.3/1 \text{ \AA}$ was based on the refined particles at 4-5 \AA resolution range, not 8/9 \AA .

More of a comment: The claim that “It is a concern in the field that whether near-atomic resolution ($<4 \text{ \AA}$) can be achieved on cell lamellas to enable the atomic modeling, and it has not yet been achieved by now” is just not true. See for example Tegunov et al. Nat Met 2021. (and others).

Thank you for pointing this. This part of the description has been removed in the previously revised manuscript.

Lastly, some rebuttal figures (e.g. 6&7) would actually be useful as supplementary figures. They should be included in the final version.

Thank you for your suggestions, we have added rebuttal figure 7 to the new revised supplementary. However, figure 6 is generated from the HSV-2 dataset and it has been published in our previous work (The Innovation, 2021), so it may be not appropriate to put figure 6 in the supplementary.

Reviewer #3 (Remarks to the Author):

I thank the authors for addressing reviewers' comments. I recommend the manuscript for publication.

Thank you for your recognition of our work.